# DAFT-GAN: Dual Affine Transformation Generative Adversarial Network for Text-Guided Image Inpainting

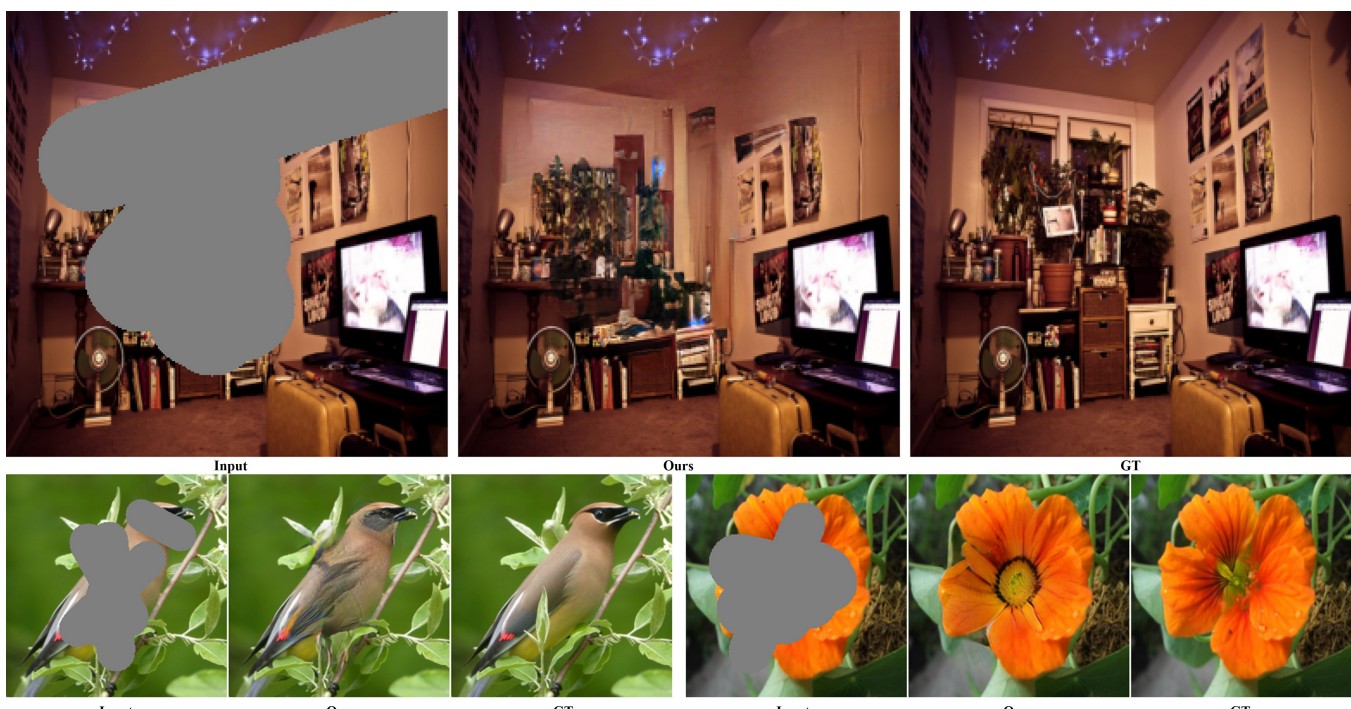

**Text (top)**: *"a room filled with furniture and accessories in a room.",* **Text (bottom left)**: *"the bird is colorful and has black eyerings a spiky tan crown and gray wings.",* **Text (bottom right)**: *"this is an orange flower with green stamen and black stripes near the ovary."*

**Figure 1: Results of proposed DAFT-GAN. Masked (left), generated (middle), and ground-truth (right) images are presented on three datasets (MS-COCO, CUB, and Oxford).**

## ABSTRACT

In recent years, there has been a significant focus on research related to text-guided image inpainting, which holds a pivotal role in the domain of multimedia processing. This has resulted in notable enhancements in the quality and performance of the generated images. However, the task remains challenging due to several constraints, such as ensuring alignment between the generated images and the accompanying text, and maintaining consistency in distribution between corrupted and uncorrupted regions, for achieving natural and fine-grained image generation. To address these challenges, previous studies developed novel architectures, inpainting techniques, or objective functions but they still lack semantic consistency between the text and generated images. In this paper, thus, we propose a dual affine transformation generative adversarial network (DAFT-GAN) to maintain the semantic consistency for text-guided inpainting. DAFT-GAN integrates two affine transformation networks to combine text and image features gradually for each decoding block. The first affine transformation network leverages global features of the text to generate coarse results, while the second affine network utilizes attention mechanisms and spatial of the text to refine the coarse results. By connecting the features generated from these dual paths through residual connections in the subsequent block, the model retains information at each scale while enhancing the quality of the generated image. Moreover, we minimize information leakage of uncorrupted features for fine-grained image generation by encoding corrupted and uncorrupted regions of the masked image separately. Through extensive experiments, we observe that our proposed model outperforms the existing models in both qualitative and quantitative assessments with three benchmark datasets (MS-COCO, CUB, and Oxford) for text-guided image inpainting.

## CCS CONCEPTS

• **Computing methodologies** → *Artificial intelligence; Computer vision; Computer vision problems; Reconstruction.*

## KEYWORDS

Text-guided image inpainting, dual affine transformation, separated mask convolution, semantic consistency

## 1 INTRODUCTION

Image inpainting [1–9] is a process that involves realistically filling in missing or corrupted areas within an image, serving a vital role in practical image processing applications such as image reconstruction, photo editing, and completing obscured regions. The core strategy in image inpainting involves predicting missing pixels by leveraging features or patterns from uncorrupted areas of the image. While these models can produce high-quality results, they face challenges in complex scenarios that demand intricate details, especially when multiple objects are involved or when the corrupted region covers a substantial portion of the image. Relying solely on surrounding pixels for generation in such cases can lead to distorted content or artifacts, prompting further research to address these limitations.

The pipeline typically follows a generative adversarial network (GAN) inpainting architecture, initially proposed in [10] and later refined by [11] to incorporate global and local aspects in the discriminator. However, addressing irregular mask holes has remained a challenge, prompting the introduction of a coarse-to-fine network [7] that includes a refinement process to enhance the generation quality. Additionally, previous studies [12, 13] proposed improved convolution techniques for feature encoding, while recent studies [14, 15] have demonstrated the benefits of utilizing fast Fourier convolutional (FFC) blocks to extract global and spatial features separately and modulate them individually for improved appearance quality. Nevertheless, accurately generating corrupted areas in complex images lacking specific patterns remains a significant challenge, as relying solely on surrounding pixels may yield semantically diverse solutions, necessitating time-consuming and resource-intensive processes to select the desired output. In specific advanced inpainting tasks, therefore, incorporating external guidance to inpainting models is necessary to control the solution space and efficiently generate desired outcomes. Since external guidance, such as lines [12], edges [16], sketches [17], or exemplars [18, 19], may offer weak visual directionality and also lack semantic context, text guidance is considered the most effective form of guidance in inpainting tasks to ensure semantic consistency. Thus, text guidance is widely used in other tasks such as text-to-image synthesis and text-guided manipulation. Text-to-image synthesis aims to generate complete images that accurately reflect the text description, while text-guided image manipulation involves modifying existing image content to align better with the text description. Inpainting tasks involve much stricter constraints compared to tasks such as text-to-image synthesis and image manipulation, as they require generating local regions of images. It is crucial to consider whether the text is accurately reflected in the generated image, if the remaining uncorrupted regions are coherent with the generated pixels, and other relevant factors.

The task of text-guided inpainting is advancing towards generating high-quality images even in more complex cases such as filling in large masked regions in an image by utilizing text descriptions as external guidance. To do this, it is crucial to effectively utilize features from the uncorrupted regions of the image. Additionally, combining the features of the image with the text features used as guidance is also vital. The attention mechanism is the most commonly used when combining features from multimodal data. In text-to-image synthesis, the attention mechanism was first proposed in AttnGAN [20], demonstrating the effectiveness of combining text features with image features. Also, a deep attentional multimodal similarity model (DAMSM) [20] was proposed for extraction of sentence and word embeddings, which are crucial features when utilizing the attention mechanism. Similarly, in this study, we utilized the pre-trained DAMSM [20] to extract sentence and word embeddings. A recurrent affine network [21] was utilized when combining sentence embeddings with images, and a network with attention added to the existing affine network was employed for refinement at the word level when combining word embeddings with images. By gradually stacking this two-path decoder block, fine-grained images that effectively reflect text were generated. While most existing studies encode features of corrupted images through a single convolution, we propose separated mask convolution blocks to distinguish between corrupted and uncorrupted regions, minimizing the information leakage of uncorrupted image features. As shown in Fig. 1, Our proposed model successfully generates corrupted regions on three benchmark datasets.

The main contributions of this paper are listed as follows.

- We propose a novel text-guided inpainting model (DAFT-GAN) for generating fine-grained images with text descriptions.
- A dual affine transformation block is proposed to incorporate visual and text features effectively in the decoder stage from a global and spatial perspective.
- A separated mask convolution block is proposed to minimize the information leakage of uncorrupted image features.
- State-of-the-art performances are achieved on three benchmark datasets (MS-COCO, CUB-200-2011, and Oxford-102).

## 2 RELATED WORK

### 2.1 Text-to-Image Synthesis

Text-to-image synthesis is a task aimed at generating complete images using text descriptions. The most crucial aspects in text-to-image synthesis are the authenticity of the generated images and the semantic consistency between the provided text description and the generated images.

The foundation of text-to-image synthesis models, such as Stack-GAN [22], utilizes a multi-stage approach to generate high-resolution images reflecting the input text. This architecture not only gradually generates high-resolution images through stages but also proposes a stacked structure to address the unstable nature of GAN models. Subsequent research [20] focused on combining text and image features, with attention mechanisms being proven effective through such studies. However, attention mechanisms also have drawbacks when generating high-resolution images since they require attention at every scale, which incurs substantial computational costs.

This computational burden can make it challenging to fully utilize text information at certain scales. To address these issues, DF-GAN [23] proposes an efficient fusion network that utilizes affine transformation networks. By combining contextual information without attention in every block, it offers significant cost advantages. While this simple and efficient network can produce images with better quality, some limitations still exist. Some regions of the images may not be recognizable or consistent with the text description at the word level. To enhance this, SSA-GAN [24] introduces weakly supervised mask predictors to guide spatial transformations. RAT-GAN [21] adds recurrent networks when connecting affine transformation blocks to address the long-term dependency problem. Through research that effectively combines text and images, the quality of image generation has been improved.

## 2.2 Image Inpainting

Image inpainting involves the reconstruction of specific uncorrupted regions within an image, serving as a critical component in various image processing applications such as photo manipulation and filling in occluded areas. Despite its importance, image inpainting remains a challenging task, with the level of difficulty being heavily influenced by the extent of the damaged regions in the image. When a significant portion of the image is corrupted, there is a higher probability of entire objects being lost from the visual context. Moreover, the absence of usable features from uncorrupted areas further complicates the restoration process. Recent advancements in deep learning models have significantly enhanced the ability to extract and leverage high-level semantic features, resulting in notable improvements in terms of the quality of generated images. Recent studies investigated different model architectures, including transformer-based [25–28], GAN-based [15, 29–32], or FFC-based [14, 15] models. In addition to model architecture, performance improvements have been achieved through the application of various inpainting techniques such as convolution [12, 13] and attention mechanisms [20].

## 2.3 Text-Guided Image Inpainting

The goal of the text-guided image inpainting task is to reconstruct corrupted images using textual and visual information to generate realistic images. MMFL [33] proposes a multimodal fusion learning approach that focuses on text descriptions corresponding to objects of interest in each image through the word demand module. Additionally, they construct it as a two-stage coarse-to-refine process to generate high-quality images. TDA [34] proposes a dual multimodal attention module that aims to achieve deeper integration by applying attention to feature maps of two inverted regions. However, all these methods attempted multimodal fusion at the encoding stage and did not utilize the provided text information during the decoding stage. This leads to an incomplete fusion of spatial details, text, and visual information. Furthermore, previous studies [33–36] utilize structures with two or multiple stages for image refinement, resulting in significant time and space consumption due to repeated encoder-decoder pairs. Therefore, we designed a network that injects text description information at the decoding stage to generate images. This strategy not only mitigates resource inefficiencies but also improves image quality through the utilization of a one-stage-dual-path architecture.

## 3 METHODS

Given a masked image $I_M = I \odot M$ (where $I$ is the original image, $M$ is given mask metric), text-guided image inpainting is the task of generating an image that aligns with the text description $t$ and maintains semantic consistency with the image. We propose DAFT-GAN, which generates images by appropriately processing text and visual features in the decoder stage.

## 3.1 Overall Architecture

As shown in Fig. 2, the proposed model consists of an encoder composed of separated mask convolution (SMC) blocks and a decoder composed of dual affine transformation (DAFT) blocks forming the Conv-U-Net structure. More specifically, the process of feature extraction involves seven SMC blocks, while text features are integrated at various resolutions through the incorporation of seven DAFT blocks to produce the final image. A comprehensive breakdown of all elements of the approach is provided as follows.

## 3.2 Separated Mask Convolution

**Seperated Mask Convolution**. SMC block performs two roles. As shown in Fig. 3, one is to distinguish the masked and the unmasked regions and conduct convolution and normalization separately on them. Another is to update the mask in a way that minimizes information leakage. SMC block takes as input a pair of the image feature $F_i^e \in \mathbb{R}^{C_i \times W_i \times H_i}$ extracted from the previous block and a mask metric $M_i \in \{0, 1\}^{W_i \times H_i}$ of the same size. The mask metric represents the mask status, where the unmasked (valid) regions are marked as 0, and the masked (invalid) regions are marked as 1. The mask metric is used to differentiate between the two regions during the encoding process. The initial $256 \times 256$ masked image and mask metric are downscaled by a factor of 2 in 6 out of the 7 blocks, excluding the first block, resulting in a final output size of $4 \times 4$.

First, from the previous block, we passed the input feature $F_i^e$ through two different convolution layers to obtain a valid feature $F_{i+1}^{val} \in \mathbb{R}^{C_{i+1} \times \frac{W_i}{2} \times \frac{H_i}{2}}$ and an invalid feature $F_{i+1}^{inval} \in \mathbb{R}^{C_{i+1} \times \frac{W_i}{2} \times \frac{H_i}{2}}$. If we use only one convolutional layer to extract features, the convolutional weights could be updated by being influenced not only by the valid region but also by the invalid region during training. Therefore, we used two convolution layers.

$$F_{i+1}^{val} = Norm(Conv^{val}(F_i^e \odot (1 - M_i))), \quad (1)$$

$$F_{i+1}^{inval} = Norm(Conv^{inval}(F_i^e \odot M_i)). \quad (2)$$

Then, we generated $F_{i+1}^e \in \mathbb{R}^{C_{i+1} \times \frac{W_i}{2} \times \frac{H_i}{2}}$ composed of $F_{i+1}^{val}$ and $F_{i+1}^{inval}$. At this point, using $M_{i+1} \in \{0, 1\}^{\frac{W_i}{2} \times \frac{H_i}{2}}$ obtained by updating $M^i$, we assigned $F_{i+1}^{val}$ to the valid region of $F_{i+1}$ and $F_{i+1}^{inval}$ to the invalid region. The mask metric update process is explained in detail as follows.

$$F_{i+1}^e = F_{i+1}^{val} \odot (1 - M_{i+1}) + F_{i+1}^{inval} \odot M_{i+1} \quad (3)$$

In $F_{i+1}^e$, the valid and invalid regions have distinctly different distributions. To prevent the loss of information in the valid region, we applied MaskNormalize, the normalization method used in [37].

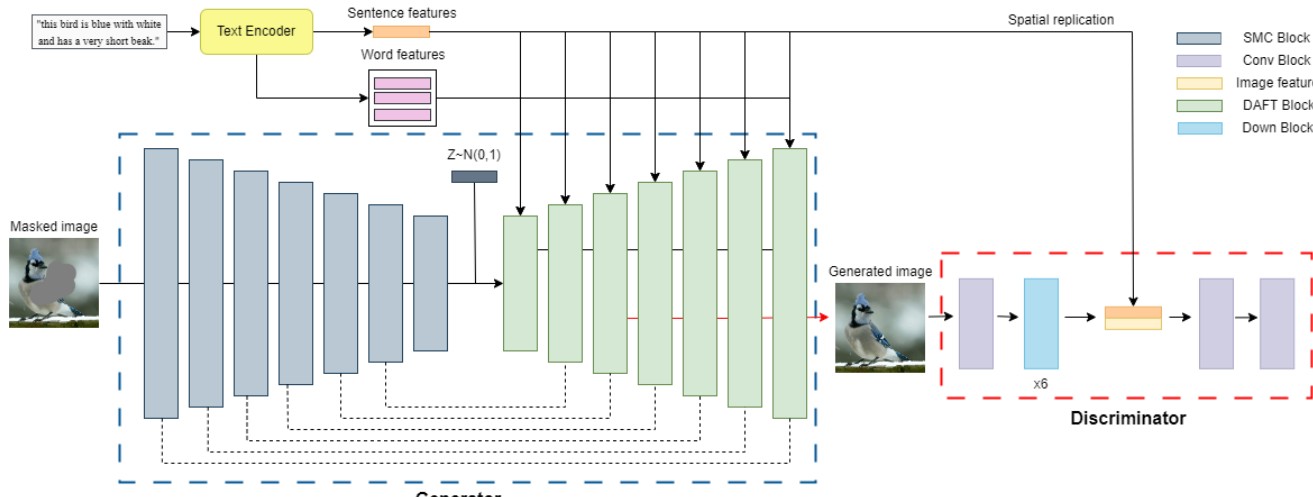

**Figure 2: Architecture of DAFT-GAN consisting of an encoder-decoder generator and a one-way discriminator. The generator extracts image features and combines them with noise and text embeddings to generate the reconstructed images.**

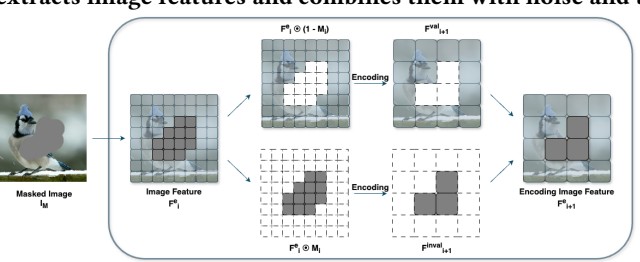

**Figure 3: Visualization of processing of the SMC block. Encoding includes convolution and normalization, generating higher-dimensional features that are downscaled by a factor of 2.**

***Mask Update****.* When performing convolution operations on image features, both the valid and invalid regions are inevitably considered at the boundary of the mask due to the receptive field. In response to this situation, we applied a mask update method that treats as a valid feature if any part of the valid region is included to minimize the loss of valid information. We used a pooling layer with the same kernel size, stride, and padding as the convolution layer to extract the image features. By doing so, when performing convolution operations on a specific region of the image feature, we can also perform pooling operations on the exact corresponding region for the mask metric. If any part of the valid region is included in the receptive field, i.e., the region in the mask metric contains both 0 and 1, we need the pooling result for that region to be 0 in order to consider the convolution result as a valid feature. Therefore, we implemented a min-pooling operation to update the mask as follows.

$$M_{i+1} = -MaxPool(-M_i) \qquad (4)$$

### 3.3 Dual Affine Transformation

***Global Fusion Path****.* As depicted in Fig. 2, the generator generates masked parts by combining the encoded feature with the U-Net structure [38]. A noise vector $z$ is sampled from a standard Gaussian distribution, passed through a fully connected layer, and then added to the encoded feature of the same dimension at the beginning of the generator. For each upsampling block, the encoded feature of the same scale was combined with the feature of the previous block through element-wise addition.

$$F_0^{gin} = MLP_1(z) \oplus F_L^e, \qquad (5)$$

$$F_i^{gin} = F_{L-i}^e \oplus F_i^g, \qquad (6)$$

where $1 \le i \le L$ ($L$ is the highest level with the smallest spatial size)

Unlike the basic U-Net [38], this model combined features using residual connections instead of channel-wise concatenation. This was done to encourage the network output to closely resemble the input during the early stages of training, which stabilizes the learning process. Additionally, residual learning generally helps the model preserve previous information at each scale and learn high-frequency contents more effectively. The input $F_i^{gin}$ passes through each recurrent affine transformation (RAT) module [21], performing the transformation for upsampling. It sequentially goes through the initial generator block and 6 upsampling blocks (the first generator block does not upsample), ultimately generating a feature map as a size of $256 \times 256$.

***Multimodal Cross Affine Transformation****.* As shown in Fig. 4, multimodal cross affine transformation (MCAT) takes the RAT output image feature maps $F_i^{gout}$ and word features $w$, hidden features $h_t$ within the same DAFT block as input, and outputs same scale image feature maps $F_i^{sout}$. The core of the MCAT module is the CrossAffine layer shown in Fig. 5. The CrossAffine layer performs cross-attention between image feature maps and word features, and obtains attention feature maps $F_{\text{spatial}} \in \mathbb{R}^{(2 \times d_w) \times H \times W}$ through channel-wise concatenation with hidden features $h_t$, and then extracts modulation parameters $\gamma_c$ and $\beta_c$ through channel-wise multi-layer perception (MLP) for affine transformation. By applying channel-wise MLP to predict modulation parameters while maintaining spatial structure and identifying where text information needs to be complemented in the current image features,

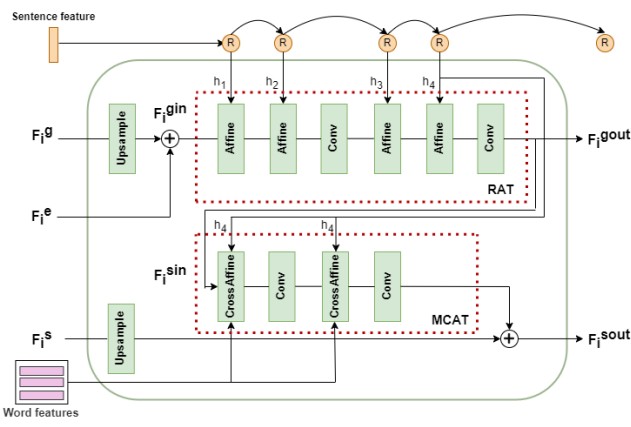

Figure 4: Structure of the DAFT block. The block is composed of RAT and MCAT modules, which respectively handle the global path and the spatial path, thereby forming a dual path architecture.

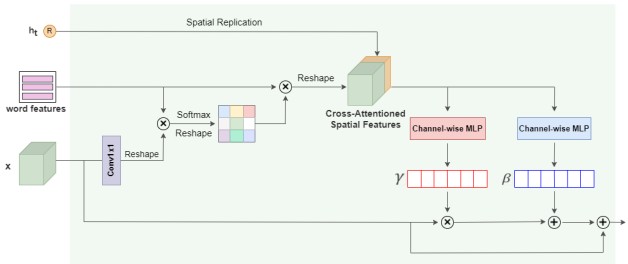

Figure 5: Diagram of CrossAffine module. The module manipulates input image features using recurrent hidden features and word features by channel wise affine transformation.

spatially refined image feature maps can be obtained. Specifically, given input image feature maps $x_{chw} \in \mathbb{R}^{C \times H \times W}$, word features $w \in \mathbb{R}^{L \times d_w}$, and hidden feature $h_t \in \mathbb{R}^{d_w}$, the image feature maps were transformed into the same semantic space as word features by a new perceptron layer $W_Q \in \mathbb{R}^{d_w \times C}$. The query $Q_F = W_Q x_{chw} \in \mathbb{R}^{d_w \times H \times W}$, and key $K_F$, value $V_F$ are word features $w$. Then, it performs cross-attention to obtain image feature maps $F_{\text{spatial}}$ for predicting modulation parameters as follows:

$$F_{\text{attn}} = softmax(Q_F K_F^T) V_F, \tag{7}$$

$$F_{\text{spatial}} = [F_{\text{attn}}; SpatialReplication(h_t)], \tag{8}$$

where SpatialReplication expands the dimensions by $H \times W$.

By combining the sentence hidden feature $h_t$ from the long short-term memory with channel-wise concatenation with $F_{\text{attn}}$, necessary global semantic information at current stage was injected, at the same time, global-spatial connection was implemented. Additionally, modulation parameters $\gamma_c$ and $\beta_c$ were predicted through a channel-wise MLP, and channel-wise affine transformation and residual connection are applied to input feature maps $x_{chw}$ to generate final output feature maps $\tilde{x}_{chw}$. This process can be represented by the following equation:

$$CrossAffine(x_{chw} \mid h_t, w) = \gamma_c x_{chw} + \beta_c, \tag{9}$$

where $\gamma_c$ and $\beta_c$ are obtained through two channel-wise MLP layers.

$$\gamma_c = ChannelwiseMLP_1(F_{\text{spatial}}), \tag{10}$$

$$\beta_c = ChannelwiseMLP_2(F_{\text{spatial}}), \tag{11}$$

$$\tilde{x}_{chw} = x_{chw} \oplus CrossAffine(x_{chw} \mid h_t, w). \tag{12}$$

In order to learn a wider range of expressions, as described in Fig. 4, CrossAffine layer and convolution layer were stacked to form a single MCAT module.

**One-Stage-Dual-Path**. Inspired by recent research on image inpainting [15], we proposed a one-stage-dual-path approach that utilizes global text features to manipulate images, focusing on a global path for manipulating global structure and a spatial path for concentrating on spatial details. As shown in Fig. 4, the global path generates a semantic-consistent global structure through the RAT module, while the spatial path leverages more semantic-consistent and fine-grained visual details through word-level and pixel-level control using the MCAT module for each scale of the output from the global path. Additionally, by upsampling the previous spatial path output and connecting it to the current output through a residual connection, the model is guided to transform only the necessary parts gradually at each step, enabling initial learning stabilization and finer control over details.

This structure resolves the time and space waste issues derived from the two-stage or multi-stage structure of the existing text-guided image inpainting [33, 35, 36], and has the advantage of generating natural images from a human evaluation perspective through organic integration and detailed role differentiation between the two paths.

### 3.4 Objective Functions

#### 3.4.1 Discriminator Objective.

**Adversarial Loss with MA-GP**. To ensure semantic consistency between the inferred image and the given text description, matching-aware zero-centered gradient penalty (MA-GP) was used [23].

$$
\begin{aligned}
L_D^{adv} = \ & E_{x \sim \mathbb{P}_{data}}[max(0, 1 - D(x, s))] \\
& + \frac{1}{2} E_{x \sim \mathbb{P}_G}[max(0, 1 + D(\hat{x}, s))] \\
& + \frac{1}{2} E_{x \sim \mathbb{P}_{data}}[max(0, 1 + D(x, \hat{s}))] \\
& + k E_{x \sim \mathbb{P}_r}[(\|\nabla_x D(x, e)\| + \|\nabla_e D(x, e)\|)^p],
\end{aligned} \tag{13}
$$

where $s$ is the given text description, $\hat{s}$ is the mismatched text description, $x$ is the actual image corresponding to $s$, and $\hat{x}$ is the generated image. $D(.)$ is the output of the discriminator, providing matching information between the image and the sentence. The $k$ and $p$ are hyperparameters of the MA-GP.

#### 3.4.2 Generator Objective.

**Reconstruction Loss**. The $\ell_1$ loss is typically optimized for an average blurry result. Despite being a non-saturating function, we incorporated perceptual loss to improve the naturalness and quality of the final image.

$$L_{rec} = \sigma_i \|\phi_i(\hat{x}) - \phi_i(x)\|_2, \tag{14}$$

**Table 1: Performance comparison of GAN-based inpainting models regarding FID, KID, PSNR, and SSIM. The evaluation was conducted on the test dataset in CUB-200-2011, Oxford-102, and MS-COCO.**

| Model | CUB-200-2011 | | Oxford-102 | | MS-COCO | |
|---|---|---|---|---|---|---|
| | FID↓ | KID↓ | FID↓ | KID↓ | FID↓ | KID↓ |
| RFR [8] | 26.09 | 1.206 | 23.11 | 1.127 | 22.59 | 1.101 |
| PDGAN [9] | 45.69 | 2.619 | 31.08 | 1.424 | 34.86 | 1.509 |
| MMFL [33] | 25.68 | 1.266 | 35.69 | 1.560 | 19.77 | 0.803 |
| TDA [34] | 13.07 | 0.394 | 19.65 | 0.681 | 11.20 | 0.586 |
| Ours | **11.33** | **0.259** | **15.75** | **0.318** | **6.59** | **0.357** |
| | PSNR↑ | SSIM↑ | PSNR↑ | SSIM↑ | PSNR↑ | SSIM↑ |
| RFR [8] | **21.28** | **0.811** | **20.76** | 0.808 | **20.82** | 0.781 |
| PDGAN [9] | 19.27 | 0.754 | 18.94 | 0.732 | 18.23 | 0.655 |
| MMFL [33] | 20.34 | 0.799 | 20.61 | **0.811** | 20.48 | 0.769 |
| TDA [34] | 20.23 | 0.797 | 19.02 | 0.753 | 19.56 | 0.652 |
| Ours | 20.46 | 0.808 | 20.25 | 0.766 | 19.64 | **0.783** |

where $\phi_i(.)$ refers to the layer activation of the pre-trained VGG-19 network.

***Adversarial Loss***. Adversarial loss is defined as follows.

$$L_G = -E_{\hat{x}}[log(D(x))], \tag{15}$$

where $\hat{x}$ is generated images.

***Text-Guided Attention Loss***. To enhance text guidance, we implemented the text-guided attention loss [33]. This method involves multiplying the attention map and the generated image $\hat{x}$ at the final scale of $256 \times 256$ with the ground-truth image $x$, and minimizing the $\ell_1$ loss of the two terms.

$$L_{attn} = \|A(w, \hat{x})\hat{x} - A(w, \hat{x})x\|_1, \tag{16}$$

where $A(.)$ performs attention and $w$ is the word features of the text corresponding to $x$.

***DAMSM Loss***. For fine-grained image-text matching that considers both sentence-level and word-level information, we adopted the DAMSM loss ($L_{DAMSM}$). Details are described in [33].

***Overall Loss***. The total loss of the generator is defined as below.

$$L_G^{adv} = \lambda_{rec} \times L_{rec} + L_G + L_{attn} + \lambda_{DAMSM} \times L_{DAMSM} \tag{17}$$

## 4 EXPERIMENTS

### 4.1 Datasets

We used the Caltech-UCSD Birds-200-2011 (CUB-200-2011), Oxford-102 Category Flower (Oxford-102), and MS-COCO datasets to train our proposed model and methods, focusing on multimodal inpainting that incorporates both text and images. While all three datasets were used to assess the performance of text-guided inpainting, the MS-COCO dataset was specifically used as a more challenging dataset for evaluating the performance of image inpainting. This is because the MS-COCO dataset contains multiple objects and relatively complex scenes compared to the other two datasets.

**Table 2: Numerical ranking scores for semantic consistency and naturalness. Semantic consistency and naturalness indicate the alignment between text and image, and the quality of the image, respectively.**

| Model | Semantic consistency | Naturalness |
|---|---|---|
| RFR [8] | 2.852 | 3.347 |
| PDGAN [9] | 3.648 | 4.213 |
| MMFL [33] | 4.008 | 4.787 |
| TDA [34] | 2.765 | 3.465 |
| Ours | **2.146** | **2.625** |
| Ground-truth | 1.134 | 1.145 |

### 4.2 Implementation Details

When training or testing our proposed model, we used randomly generated masks in irregular shapes. By doing so, we can construct a more robust model for inference and demonstrate the ability to generate realistic images even when large corrupted regions are present in the image. Subsequently, we evaluated the performance of the proposed model through several experiments. We confirmed the effectiveness and competitiveness of our proposed model by conducting quantitative in Table 1 and qualitative evaluations in Table 2 with the other models proposed in previous studies. After that, to validate the effectiveness and performance improvement of each method in the final model we selected, we sequentially evaluated the application of methods from the baseline model as shown in Table 3. We utilized a training approach with diverse irregular masks. We conducted the experiment on an NVIDIA RTX3090 Ti GPU. The parameters of the generator in the model were optimized with a learning rate of $10^{-4}$, while the parameters of the discriminator were optimized with a learning rate of $4 \times 10^{-4}$, both using the Adam optimizer. The sentence embedding and word embedding utilized in this study were extracted using a pre-trained text encoder from the DAMSM proposed in AttnGAN [20]. The final objective function weights were set as $\lambda_{rec} = 0.2$ and $\lambda_{DAMSM} = 0.01$

### 4.3 Quantitative Results

The image inpainting task allows for evaluating the quality of generated images by comparing how well the corrupted parts are reconstructed to the ground truth. To assess the quality of generated images, the Fréchet inception distance (FID) [39] and the kernel inception distance (KID) [40] were used to measure the distribution between ground-truth and generated images. When evaluating reconstruction, the peak signal-to-noise ratio (PSNR) and the structural similarity index (SSIM) [41] were used to measure the difference between generated images and ground-truth at the pixel level. Lower values for FID and KID, and higher values for PSNR and SSIM indicate better performance. As shown in Table 1, we observed that our proposed model has a significant impact on integrating text and achieving high scores compared to the existing models on all three datasets, with a particularly significant improvement seen on the MS-COCO dataset. Despite being relatively challenging due to its inclusion of multiple objects and complex scenes, the significant improvement on the MS-COCO dataset indicates effective semantic matching between text and images. Despite the

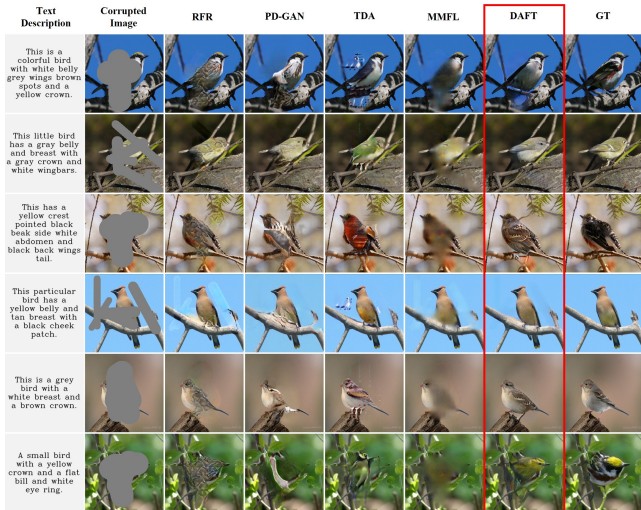

**Figure 6: Qualitative comparison of ours with other models on the CUB-200-2011 dataset.**

**Table 3: Performance comparison of each component added. To compare three different affine transformation methods, the baseline (DF-GAN) was compared against methods that applied the SSA and RAT affine transformations. DAFT employed the RAT affine transformation.**

| Methods | FID↓ | KID↓ | PSNR↑ | SSIM↑ |
|---|---|---|---|---|
| Our Baseline (DF-GAN) | 19.31 | 0.759 | 18.15 | 0.790 |
| Our Baseline + SSA | 18.79 | 0.730 | 19.80 | 0.796 |
| Our Baseline + RAT | 17.25 | 0.593 | 19.87 | 0.806 |
| Our Baseline + MCAT | 17.84 | 0.641 | 20.13 | **0.814** |
| Our Baseline + DAFT (MCAT + RAT) | 14.23 | 0.418 | 18.95 | 0.757 |
| Our Baseline + DAFT + SMC | 12.57 | 0.293 | 19.06 | 0.742 |
| Our Final (Diverse Mask + SMC + DAFT) | **11.33** | **0.259** | **20.46** | 0.808 |

significant improvements in FID and KID scores across all datasets, the improvements in PSNR and SSIM are relatively modest. This phenomenon may be explained by the distinction that PSNR and SSIM metrics evaluate distances at the pixel level, while our model operates on feature-level distances to produce images that exhibit a more realistic appearance. Consequently, the improvement in reconstruction metrics compared to previous models that rely solely on pixel-level distances might be lower. Nonetheless, it is evident that in some cases, other models achieve similar or even higher reconstruction scores.

## 4.4 Qualitative Results

In the context of generative models, qualitative evaluation from a human perspective is also important. In the text-guided image inpainting task, the most commonly used metrics for qualitative evaluation are semantic consistency and naturalness. Semantic consistency measures how well the generated image matches the given text, while naturalness measures how natural the generated image appears. We conducted a human evaluation with 15 volunteers with 100 randomly generated images, ranging from rank 1 (high quality) to rank 5 (low quality), to obtain scores for semantic consistency and naturalness. As scored in Table 2, we observed that our model

**Table 4: Evaluations of different Mask Ratios on CUB-200-2011 datasets with diverse irregular masks.**

| Model | 20%-50% | | 20%-30% | | 30%-40% | | 40%-50% | |
|---|---|---|---|---|---|---|---|---|
| | FID↓ | KID↓ | FID↓ | KID↓ | FID↓ | KID↓ | FID↓ | KID↓ |
| RFR [8] | 26.09 | 1.206 | 16.13 | 0.624 | 28.79 | 1.404 | 42.32 | 2.416 |
| PDGAN [9] | 45.69 | 2.619 | 29.31 | 1.439 | 46.52 | 2.722 | 66.09 | 4.416 |
| MMFL [33] | 25.68 | 1.266 | 15.92 | 0.682 | 27.51 | 1.424 | 41.60 | 2.470 |
| TDA [34] | 13.07 | 0.394 | 10.20 | 0.259 | 13.30 | 0.399 | 16.67 | 0.598 |
| Ours | **11.33** | **0.259** | **8.815** | **0.159** | **12.01** | **0.271** | **15.28** | **0.432** |
| | PSNR↑ | SSIM↑ | PSNR↑ | SSIM↑ | PSNR↑ | SSIM↑ | PSNR↑ | SSIM↑ |
| RFR [8] | **21.28** | **0.811** | 22.72 | 0.856 | 20.80 | 0.801 | **19.42** | **0.745** |
| PDGAN [9] | 19.27 | 0.754 | 20.86 | 0.795 | 18.91 | 0.747 | 17.52 | 0.700 |
| MMFL [33] | 20.44 | 0.807 | **22.83** | 0.859 | **21.19** | 0.795 | 19.24 | 0.738 |
| TDA [34] | 20.23 | 0.797 | 21.52 | 0.845 | 19.83 | 0.787 | 18.61 | 0.732 |
| Ours | 20.46 | 0.808 | 22.33 | **0.861** | 20.08 | **0.802** | 18.51 | 0.742 |

not only outperforms the other models [8, 9, 33, 34] in quantitative evaluation but also achieves the best scores in qualitative evaluation. As evident from the metrics in Fig. 6, while other models generated awkward images with blurriness or artifacts, our model generated natural-looking images without any anomalies compared to the ground truth. To ensure high-quality image generation across all datasets, we conducted experiments with various text descriptions using our proposed model. In Fig. 7, we confirmed the generation of photorealistic images on the CUB and the Oxford datasets. Particularly in Fig. 8, despite the diverse object types and more complex scenes, our model effectively incorporated the text descriptions to generate remarkably natural images, even for the challenging MS-COCO dataset.

## 4.5 Effectiveness of Each Proposed Method

In Table 3, we conducted an ablation study on the CUB-200-2011 dataset to assess the performance changes resulting from sequentially applying our proposed methods to the baseline model. The baseline model utilizes a basic convolutional block to encode the features of the masked image and employs only one affine transformation network [23] along with sentence features. From this baseline, we measured the performance changes when adding the weakly supervised mask predictor proposed in [24], and when adding the recurrent affine network proposed in [21]. Then, we evaluated the performance when adding cross refinement, which involves using two affine transformation networks to refine by integrating word features. Additionally, we evaluated the performance by incorporating the DAFT Block, which connects the features of each encoder to the decoder using residual connections. Then, we examined the impact of replacing the basic block with the SMC block in our final model. Finally, to build a robust inpainting model that is independent of the shape of corrupted masks, we trained our model using various irregular masks. By sequentially applying our proposed methods, we confirmed significant improvements in performance based on the evaluation metrics.

## 4.6 Text Controllability for Image Manipulation

Through text guidance, we can manipulate specific parts to create desired images or generate diverse images that are semantically consistent with various text descriptions. In other words, it is possible to extend text-guided image manipulation using our trained model. We confirmed it through the following process. First, mask

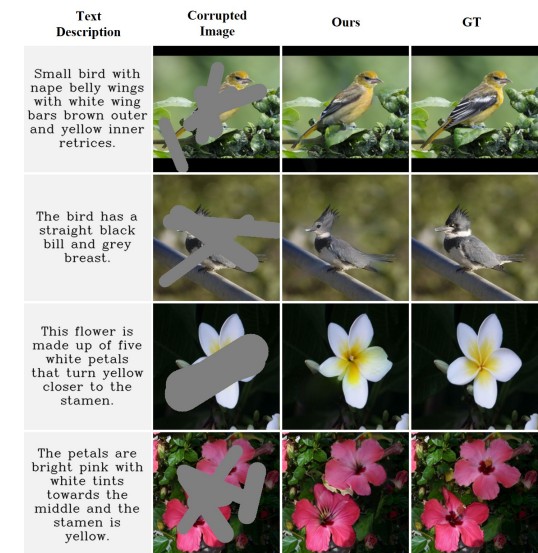

**Figure 7: Results of our proposed model on CUB-200-2011 (first and second rows) and Oxford-102 (third and fourth rows) datasets. Corrupted (left), generated (middle), and ground-truth (right) images are presented.**

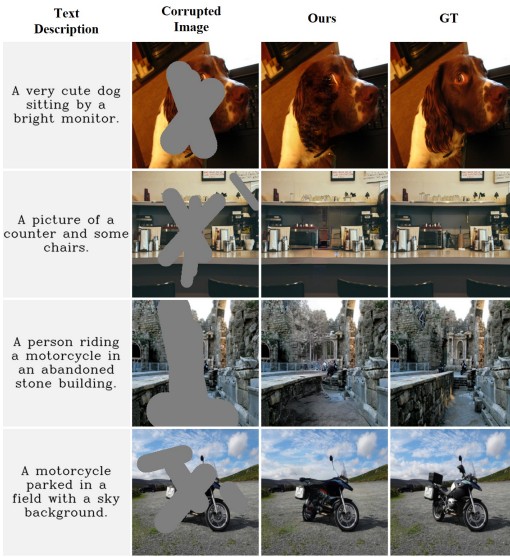

**Figure 8: Results of our proposed model on MS-COCO Dataset. Corrupted (left), generated (middle), and ground-truth (right) images are presented.**

the part of the image we want to manipulate. Second, provide a textual description of the anticipated image post-manipulation. Third, feed the masked image and text into the model.

We presented two different results when two different sentences were inserted into each image. As shown in Fig. 9, the first and second examples utilize different images and masks to show color and size variations, as well as the inclusion of relevant semantic information in generated images. The results show the ability to

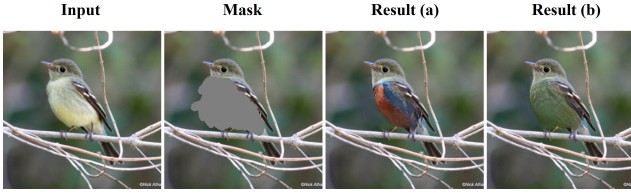

**Text (a)**: *"A thin bird with blue and red breast."*
**Text (b)**: *"A fat bird with green breast."*

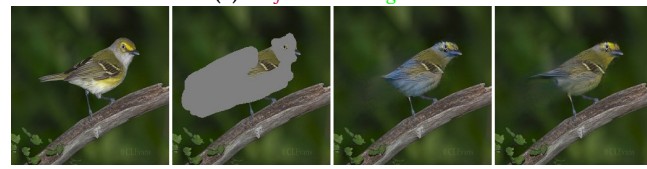

**Text (a)**: *"A tiny bird with blue heads and short blue tail."*
**Text (b)**: *"A tiny bird with yellow heads and long black tail."*

**Figure 9: Generatated images with different text guidance.**

accurately locate and generate modifications based on the provided text description. This experiment shows that it is possible to generate natural images that are semantically consistent with any arbitrary mask, and it indicates the potential for expansion into image manipulation.

## 4.7 Analysis of Performance across Varied Mask Ratios

Our goal is to develop a robust model capable of effectively reconstructing images corrupted by various sizes and shapes encountered in real-world scenarios. Therefore, we evaluated the performance of irregular random masks across different ratio ranges 20%-30%, 30%-40%, 40%-50%, and 20%-50%. As shown in Table 4, the FID and KID metrics steadily worsen as the mask ratio increases from 20% to 50%. This is an unavoidable result, as the available clues for predicting the corrupted regions diminish as the mask size grows larger. However, TDA and our model demonstrated relatively robust performance and ours achieved the best performance across all mask ratio ranges on the FID and KID metrics.

## 5 CONCLUSIONS

In this study, we propose a novel model (DAFT-GAN) to address the current challenges for text-guided image inpainting. Our approach involves the integration of two affine transformation networks to progressively incorporate text and image features, thereby enhancing the semantic consistency between the generated images and associated text descriptions. Using global text features, our model initially generates coarse results, which are subsequently refined using spatial details, leading to an overall improvement in the quality of the generated images. Additionally, we propose SMC blocks to reduce information leakage of uncorrupted features by encoding corrupted and uncorrupted regions of the masked image separately. Strikingly, we demonstrate that our proposed method outperforms existing methods in terms of both qualitative and quantitative evaluations. Our contributions represent significant advancements in text-guided image inpainting and pave the way for further research in the field of text-guided image inpainting.

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
