# OpenReview forum: "DAFT-GAN: Dual Affine Transformation Generative Adversarial Network for Text-Guided Image Inpainting"
_acmmm.org/ACMMM/2024/Conference — MM2024 Poster_

### Official Review · Reviewer_HyB9 · 2024-05-13

**Rating:** 4
**Confidence:** 4

**Summary:**

The paper presents DAFT-GAN, a new model designed to enhance text-guided image inpainting by ensuring semantic consistency between text descriptions and generated images. DAFT-GAN uniquely uses two affine transformation networks for a gradual merger of text and visual features, delivering refined image quality. The model advances existing methods with innovative features: a recurrent affine network for integrating sentence embeddings, an attention-equipped network for word-level refinement, and separate encoding of damaged versus undamaged image regions. The proposed DAFT-GAN model demonstrates superior performance over current models on MS-COCO, CUB-200-2011, and Oxford-102 datasets, which underscores its effectiveness in generating fine-grained, textually consistent images.

**Strengths:**

1. The innovative incorporation of two affine transformation networks allows for a gradual combination of textual and visual features, significantly improving the semantic consistency between text descriptions and generated images.
2. Demonstrating state-of-the-art performance on three notable benchmark datasets (MS-COCO, CUB-200-2011, and Oxford-102) underscores the model’s effectiveness and superiority over existing image inpainting methods.
3. The model uniquely leverages both global features from the text for initial coarse generation and spatial text features through attention mechanisms for refined output, ensuring the generated images closely align with the textual input.
4. The introduction of separated mask convolution blocks to differentiate between corrupted and uncorrupted areas offers a novel approach to handling image inpainting, setting the research apart from conventional single convolution methods.

**Limitations:**

1. Image and Citation Placement: he current arrangement of images and their corresponding citations is not optimal, leading to potential confusion and disruption in the reader's engagement with the main text.
2. Analytical Depth and Insight: More analyses and insight are needed in the experiments. Some metrics reported do not outperform those in previous literature, yet the paper lacks a comprehensive discussion on the possible underlying reasons or the implications of these outcomes. A more in-depth analysis could provide valuable insights, explore potential limitations of the current methodology, and suggest areas for future improvement.
3. The discussion regarding the calculation of the loss function, while technically sound, has been misplaced in the main content of the paper. Considering that this is a widely recognized standard practice, it would be more fitting to include such discussions in the background section of the paper.
4. Some of the formulas are not clearly explained. Specifically, the explanation surrounding Equation (14) lacks clarity regarding the meaning and application of the expression for /sigma_i. This lack of clarity can lead to confusion among readers.

**Suitability:**

3

---

### Official Review · Reviewer_k48b · 2024-05-20

**Rating:** 2
**Confidence:** 3

**Summary:**

This paper focuses on text-guided image inpainting, and proposes a dual affine transformation generative adversarial network (DAFT-GAN).   Specifically, DAFT-GAN introduces two affine transformation networks to combine text and image features gradually.  The first one leverages text global features generate coarse results. The second one utilizes attention mechanisms and spatial of the text to refine the coarse results. The output reults of these dual paths are connected by residual connections.

**Strengths:**

The experimental results are sufficient, all the proposed modules are well-ablated. The paper is well-organized and esay to follow.

**Limitations:**

Considering the rapid development of the field of image generation and editing, the baselines compared in the paper are too old. For example, TDA is proposed in  2020, MMFL is proposed in 2020, PDGAN is proposed in 2021, which significantly decrease the value of the paper at current time. I also recommend the authors to compare with Stable Diffusion based inpainting methods.

**Suitability:**

3

---

### Official Review · Reviewer_P4hr · 2024-05-24

**Rating:** 3
**Confidence:** 3

**Summary:**

This paper proposes a Dual Affine Transformation Generative Adversarial Network to maintain the semantic consistency for
text-guided inpainting. DAFT-GAN integrates two affine transformation networks to combine text and image features gradually for each decoding block. The experiments on three benchmark datasets show its effectiveness.

**Strengths:**

1. This paper is well-written and the method is easy to follow.

2. This paper conducts comprehensive experiments by employing multiple metrics to evaluate the model's performance on three benchmarks.

**Limitations:**

The major concern with this paper is its inadequate comparative methodology. The experimental section lacks a comparison with the latest methods that have been proposed in recent years, which hinders the paper's ability to substantiate its claim of "outperforming existing models." There have been numerous advancements in text-guided image inpainting based on diffusion, and while it may be unfair to compare the authors' small model with these pretrained models, it is essential for the authors to introduce and discuss them in the related work section to ensure the paper's timeliness and comprehensiveness. I recommend that the authors focus on highlighting the advantages of their own lightweight and fast-generating models when comparing them to diffusion models, emphasizing the aspects of model efficiency and generation speed. This will provide a clearer understanding of the strengths of their small model when compared to diffusion models.

**Suitability:**

3

---

### Meta-Review · Area_Chair_aeP8 · 2024-06-26

**Recommendation:** Accept (Poster)
**Confidence:** 5

**Metareview:**

This paper initially received mixed scores, including one "borderline reject," one "weak reject," and one "borderline accept." After the rebuttal, the reviewers were very satisfied with the additional experimental results and related explanations, and all reviewers raised their scores. Ultimately, this paper received two "borderline accept" and one "weak accept" scores. Based on this, I recommend accepting this paper. I hope the authors can include the additional results from the rebuttal in the final version of the paper, as requested by the reviewers.